# A One Health assessment of antimicrobial-resistant Enterobacterales in migratory little stints (*Calidris minuta*) and aquatic ecosystems in the Kenyan Rift Valley

Catherine W. Mbuthia[1,2,3,4*], Rael J. Too[2], Alexandra Mzula[1], Titus S. Imboma[5], John Kiiru[2], Samuel Kariuki[2], Abubakar S. Hoza[1]

**1** Department of Veterinary Microbiology, Parasitology and Biotechnology, College of Veterinary Medicine and Biomedical Sciences, Sokoine University of Agriculture, Morogoro, Tanzania, **2** Centre for Microbiology Research, Kenya Medical Research Institute, Nairobi, Kenya, **3** SACIDS Foundation for One Health, Sokoine University of Agriculture, Morogoro, Tanzania, **4** Department of Biological and Physical Sciences, School of Pure and Applied Sciences, Karatina University, Karatina, Kenya, **5** Ornithology Section, Zoology Department, National Museums of Kenya, Nairobi, Kenya

* katieymbuthia@gmail.com

## Abstract

Palearctic migratory little stints (*Calidris minuta*) can acquire resistant bacteria from contaminated environments and facilitate their transboundary spread. This two-year repeated cross-sectional study assessed the frequency and distribution of multidrug-resistant (MDR) and extended-spectrum beta-lactamase (ESBL)-producing Enterobacterales. Isolates were recovered from fecal samples of *C. minuta* foraging at the shores of Lakes Bogoria (low anthropogenic activities) and Magadi (high anthropogenic activities), as well as from peripheral freshwater sources shared by birds, humans, livestock and wildlife. A total of 184 fecal samples and 48 water samples were collected upon the birds' arrival from the Arctic (cohort 1) and predeparture from the Rift Valley lakes (cohort 2). Samples were cultured, bacterial isolates were identified using MALDI-TOF MS platform and tested against 12 antimicrobials using the Kirby-Bauer disk method. Of the 294 isolates (16 genera and 33 species), *Enterobacter* spp (31.0%, n = 91) and *Escherichia coli* (17.3%, n = 51) predominated. Resistance was highest for ampicillin (50%) and lowest for meropenem (1.0%). The predominant MDR phenotype was a combination of resistances to ampicillin, tetracycline, and sulfamethoxazole-trimethoprim. Specifically, 38 (12.9%) isolates were MDR, 37 (12.6%) co-expressed ESBL-MDR traits, and 19 (6.5%) were ESBL producers that did not meet MDR criteria. *Enterobacter* spp showed the highest frequencies of MDR (8.2%, n = 24) and combined ESBL-MDR (4.8%, n = 14) phenotypes, while *Acinetobacter* spp (3.4%, n = 10) were the most frequent ESBL producers. The statistically non-significant differences (p > 0.05) across study areas, sample sources, and cohorts suggest that resistant strains are pervasive throughout these

**Data availability statement:** All relevant data are within the paper and its Supporting Information files.

**Funding:** This research was supported by the Regional Scholarship and Innovation Fund (RSIF) of the Partnership for Skills in Applied Sciences, Engineering and Technology (PASET) (Project Grant No. P165581) grant to the SACIDS Foundation for One Health at the Sokoine University of Agriculture (SUA). Funder website: https://www.rsif-paset.org/grants-scholarships/. Catherine W. Mbuthia is a recipient of the RSIF-PASET doctoral scholarship at SUA. The funders had no role in study design, data collection, analysis, and the decision to publish, or preparation of the manuscript.

**Competing interests:** The authors have declared that no competing interests exist.

landscapes, irrespective of anthropogenic pressures. This is the first study to link *C. minuta* to the antimicrobial resistance (AMR) epidemiological circuit. Our findings underscore the need to include migratory wild birds in AMR surveillance and utilizing whole-genome sequencing to accurately trace the origin and dissemination pathways of AMR strains.

## Introduction

Antimicrobial resistance (AMR) poses a critical and multi-faceted threat to global health, with impacts spanning human, livestock, wildlife, and environmental domains [1]. The pervasive detection of AMR at the humans, animals, and the environment interfaces underscores the necessity of the One Health framework for mounting an effective response [2,3]. Although the natural environment serves as the primordial reservoir for AMR [4], contemporary acceleration of the resistome is primarily driven by anthropogenic activities [5]. The systematic misuse of antimicrobials in clinical, veterinary and agriculture settings, compounded by industrial discharge creates persistent selective pressures [6–10]. These human-derived pressures culminate in significant environmental pollution, effectively transforming ecosystems into conduits for resistant bacteria and their genes. This ecological contamination spills over into wildlife populations; notably, migratory wild birds serve as bioindicators of this phenomenon, exhibiting high rates of AMR strains acquired through colonization of contaminated habitats [9,11,12].

Within the One Health context, migratory birds are increasingly recognized as potential global vectors and sentinels for AMR [13]. Their long-distance movements across continents [12,14,15] bring them into contact with a spectrum of anthropogenically influenced landscapes, from farmlands [9], water bodies [16,17], to landfills [18] and urban centers [19], which serve as hotspots for the exchange of resistant bacteria. The little stint (*Calidris minuta*) exemplifies this phenomenon. With a global population of approximately 1.6 million [20], this small shorebird undertakes a vast annual migration from its Arctic breeding grounds to wintering sites across Africa and Asia [20,21]. During its residency in the Rift Valley lakes of Kenya, the species occupies a specific ecological niche: it forages for invertebrates, specifically chironomids (non-biting midges) and ephydrids (alkali flies) on the mudflats of lake shores, including the saline-alkaline lakes of Bogoria and Magadi, while relying on nearby freshwater sources such as rivers, springs, streams and artesian boreholes for hydration. Because these freshwater points are communal hubs shared by humans, livestock and wildlife, they serve as critical interfaces for the acquisition and dissemination of resistant bacteria [22].

The public health significance of this dynamic is heightened by the specific bacteria being isolated. The detection of resistant Enterobacterales and notorious ESKAPE pathogens in wild birds [23–25] is particularly alarming, as these groups are frequently associated with multidrug resistance (MDR) including resistance to the World Health Organization (WHO) critically important antimicrobials (WHO-CIA) List for human medicine such as third- to fifth-generation cephalosporins and carbapenems in clinical settings. Resistance to clinical last-line agents like third to fifth generation

cephalosporins and carbapenems is often mediated by genes for extended-spectrum beta-lactamases (ESBLs) and carbapenemases. Critically, these genes are commonly carried on mobile genetic elements, facilitating their dissemination across bacterial species and thereby intensifying the global AMR crisis [26].

While anthropogenic activities are recognized as a key driver of AMR dissemination by migratory birds [17,19,27], data from Kenya on this phenomenon remain scarce. This study aimed to address this critical gap by evaluating the influence of anthropogenic activity and consequently environmental pollution on phenotypic ESBLs and MDR frequencies and distribution in the little stint, *Calidris minuta,* thereby assessing its potential as a bioindicator of environmental AMR contamination. To achieve this, we compared the resistance profiles of Enterobacterales isolated from both *C. minuta* and the surrounding foraging and aquatic sources that were characterized by differing levels of anthropogenic pressures.

## Materials and methods

### Study design and study area

This was a two-year repeated cross-sectional study conducted in two distinct cohorts. The first cohort from *C. minuta* and water samples was collected in October 2021, coinciding with the arrival of *C. minuta* at the lakes from the Arctic. The second phase of sample collection took place in April 2022, during their pre-departure from the lakes back to the Arctic. The study was conducted along the shores of Lake Magadi (02° 05′ 46″ S, 36° 15′ 32″ E) and Lake Bogoria (02° 21′ 18″ N, 36° 4′ 0″ E) and other surrounding aquatic sources (Figs 1 and 2). Both lakes are endorheic saline-alkaline, located in the arid Kenyan Rift Valley and serve as crucial wintering grounds for *C. minuta*. Lake Magadi, a concession lake located in the southern Rift Valley near the Tanzanian border, is characterized by high concentrations of sodium chloride and carbonate (pH 11.2). Anthropogenic activity there is substantial due to the commercial trona and soda ash mining industry. Furthermore, local livestock-keepers, the Maasai and tourists frequently utilize the relatively hot (33–45°C) spring pools for their purported therapeutic benefits.

In contrast, Lake Bogoria is a national reserve in the northern Rift Valley, with comparatively lower concentrations of sodium chloride and carbonate (pH 10.2). Its hot springs are a major tourist attraction but typically maxes out at approximately 98.5°C to 100°C, precluding human contact. Since the main bodies of both lakes are unsuitable for hydration or domestic use due to their extreme chemistry, significant interaction between humans, livestock, and *C. minuta* occurs at the surrounding peripheral freshwater sources, including springs, rivers, marshes, and boreholes. These two sites were selected for their contrasting levels of anthropogenic impact: Lake Magadi represents a high-activity industrial and residential site, while Lake Bogoria serves as a relatively pristine comparative environment.

### Sampling strategy, sample collection, handling, and transportation

**Sample collection from *C. minuta*.**  A purposive sampling technique was employed to collect fecal or cloacal samples from *C. minuta*. Birds were captured using mist nets positioned at strategic foraging and hydration sites along the lakeshores and shared peripheral freshwater sources (springs, rivers, marshes, and boreholes). To prevent resampling, each bird was fitted with a uniquely coded steel ring and comprehensive biometric data (age, weight, fat score, moult stage) were recorded. Samples were collected swiftly by experienced ornithologists to minimize handling time and bird stress. Cloacal and fecal samples were immediately preserved in Cary-Blair transport medium (Oxoid, Basingstoke, UK) and maintained at −4 °C in the field and transported to the laboratory for processing within 72 h of collection.

**Water sample collection.**  Water samples were collected using criterion purposive sampling at *C. minuta* lakeshore foraging and trapping locations, as well as the shared peripheral freshwater sources, to characterize the epidemiological interface between wildlife, livestock, and humans. Approximately, 50 mL of water was collected from each site and stored in sterile 50 mL Falcon tubes at −20 °C until transport. All samples were processed within 72 h of collection.

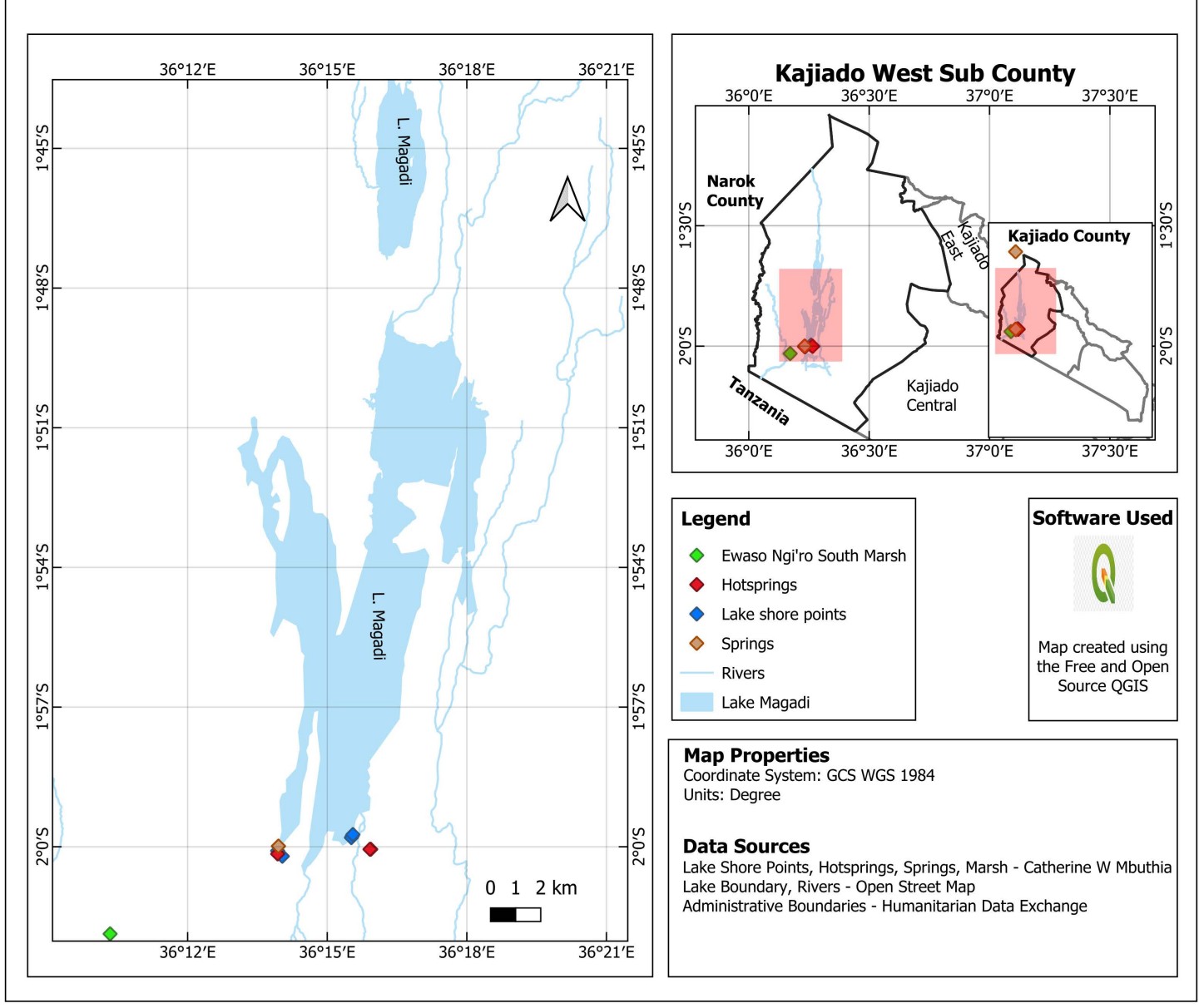

**Fig 1. Geographic distribution of Lake Magadi and other aquatic sampling points.** >Map created by the authors using QGIS version 3.4. Administrative boundaries are sourced from the Humanitarian Data Exchange (licensed under CC BY 4.0). Lake boundaries and river networks are sourced from OpenStreetMap (available under the Open Database License). All other data points were collected by the authors. This figure is original and licensed under CC BY 4.0.

## Sample processing and laboratory procedures

**Fecal and cloacal samples from *C. minuta*.** Samples were enriched in buffered peptone water (BPW) at 37 °C for 18–24 h, post-enrichment, aliquots were sub cultured onto both MacConkey (MAC) and Eosin Methylene Blue (EMB) agars (Oxoid, Basingstoke, UK) for the isolation of Enterobacterales.

**Water samples.** Water aliquots (50 mL) were filtered through multiple cellulose membrane filters (Thermo Fischer Scientific™ Nalgene™ Filter Membranes; Fisher Scientific, Hampton NH) using a Millipore vacuum filtration system. Due

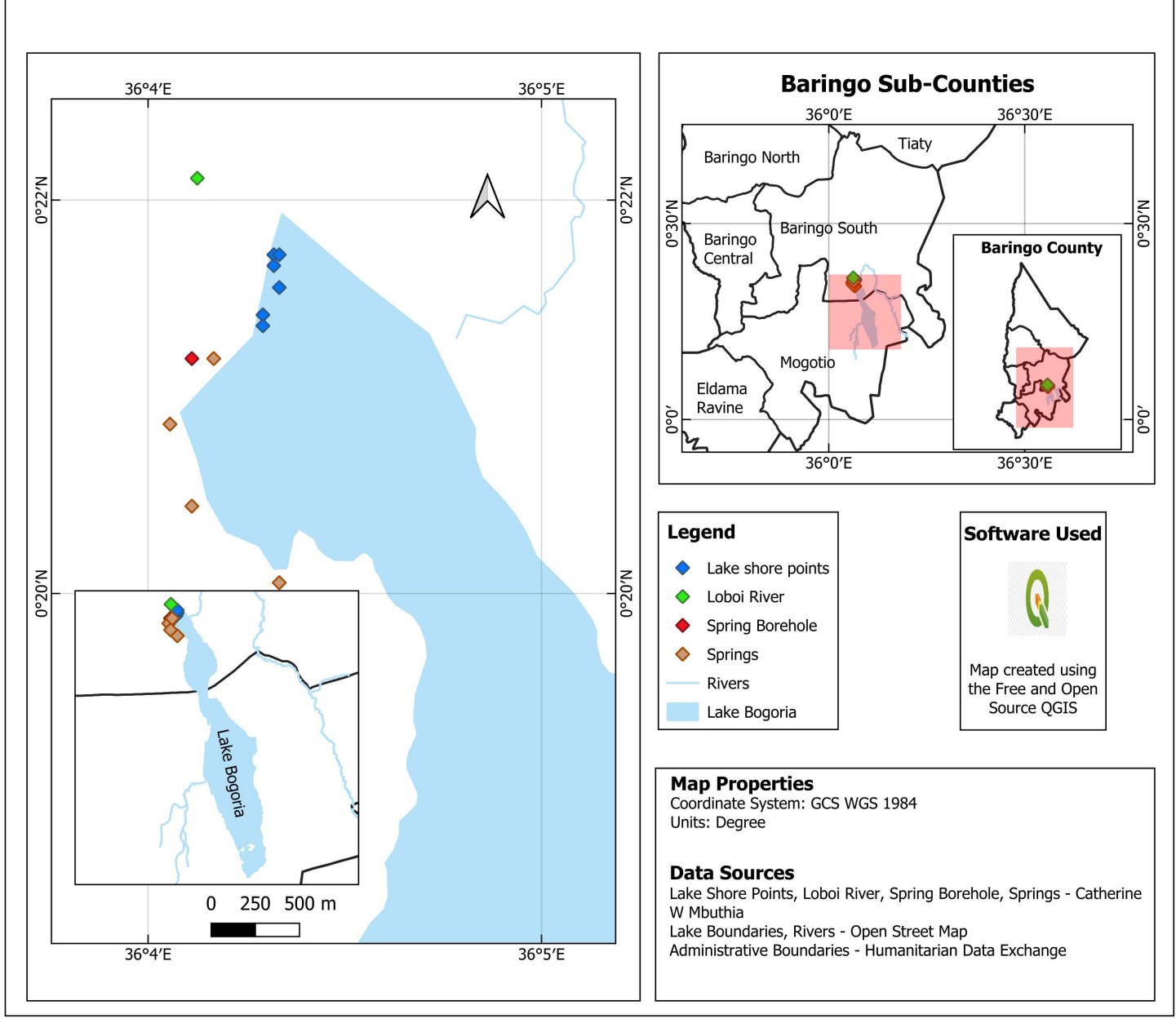

**Fig 2. Geographic distribution of Lake Bogoria and other aquatic sampling points.** > Map created by the authors using QGIS version 3.4. Administrative boundaries are sourced from the Humanitarian Data Exchange (licensed under CC BY 4.0). Lake boundaries and river networks are sourced from OpenStreetMap (available under the Open Database License). All other data points were collected by the authors. This figure is original and licensed under CC BY 4.0.

to high turbidity in some water samples, a serial filtration approach was employed to prevent membrane clogging and ensure maximum bacterial recovery. Filters of decreasing pore sizes (41μm, 20 μm, 10 μm, 1.2 μm, 0.8 μm, and 0.45 μm) were used. For each sample, all membranes were aseptically transferred into 50 mL MAC broth within sterile Whirl-Pak® bags, mixed thoroughly, and incubated at 37 °C overnight [28]. Following enrichment, the broth was sub cultured onto MAC and EMB agar plates.

**Identification and storage.** The MAC and EMB plates were incubated at 37 °C for 18–24 h. Suspected colonies were selected based on lactose fermentation profiles categorized as lactose fermenters (pink colonies), late lactose fermenters (pale pink), and non-lactose fermenters (colorless), as well as morphological features targeting Enterobacterales. A single pure colony from each suspect isolate was definitively identified using Matrix-Assisted Laser Desorption/Ionization Time of Flight Mass Spectrometry (MALDI-TOF MS) (BRUKER, Bremen, Germany) in accordance with the manufacturer's protocol. Confirmed isolates were preserved in tryptic soy broth supplemented with 15% glycerol and stored at −80 °C for subsequent analysis.

**Phenotypic antimicrobial susceptibility testing.** Antimicrobial susceptibility was determined using the Kirby-Bauer disk diffusion method, following the Clinical and Laboratory Standards Institute guidelines [29]. Previously identified isolates were recovered from frozen stock, sub cultured on MAC agar, and then grown on Mueller-Hinton agar (MHA) (Oxoid, Basingstoke, UK). Bacterial suspensions were adjusted to a 0.5 McFarland turbidity standard in sterile saline and uniformly lawned onto MHA plates.

A panel of 12 antimicrobial disks (Oxoid, Basingstoke, UK), representing 8 drug classes of varying concentrations (μg) and relevant to human and veterinary medicine and recommended for Enterobacteriaceae were used. The panel included; ampicillin (AMP, 10 μg), amoxicillin-clavulanic acid (AMC, 30 μg), cefepime (FEP, 30 μg), cefotaxime (CTX, 30 μg), ceftriaxone (CRO, 30 μg), cefuroxime (CXM, 30 μg), meropenem (MEM, 10 μg), ciprofloxacin (CIP, 5 μg), tetracycline (TE, 30 μg), chloramphenicol (C, 30 μg), gentamicin (CN, 10 μg), sulfamethoxazole-trimethoprim (SXT, 25 μg). Following incubation at 37 °C for 18–24 h, zones of inhibition were measured and interpreted as susceptible, intermediate or resistant according to CLSI 2024 breakpoints.

**Phenotypic detection of ESBL and MDR.** ESBL production was screened using the double-disk synergy test (DDST) [29]. Isolates displaying characteristic synergy zones between AMC and third-generation cephalosporins (cefotaxime or ceftriaxone) were identified as ESBL producers. MDR was defined as non-susceptibility to at least one agent in three or more antimicrobial categories [30]. Quality control was ensured using the *E. coli* ATCC 25922 reference strain.

## Statistical analyses

Comparative analyses were performed to assess the occurrence of resistant bacterial isolates across four primary variables: (i) Study areas (Lake Magadi vs Lake Bogoria), (ii) Sample sources (*C. minuta* vs water samples) (iii) Cohorts (Cohort 1 vs Cohort 2) and (iv) Antimicrobial agents. Normality was assessed using the Shapiro-Wilk test on the residuals of resistance rates. The test yielded a significant result (p-value = 0.0046), indicating that that the data were non-normally distributed. Consequently, a tie-corrected Kruskal-Wallis test was employed to evaluate differences in median resistance rates across the variables. All statistical tests were performed using Stata version 17. 0, and all tests of hypotheses were conducted at significance level of α = 0.05.

## Ethical considerations

The study obtained scientific and ethical clearances from the National Commission for Science Technology, and Innovation (NACOSTI, ID: 333849, License No: NACOSTI/P/21/13520) and the Directorate of Postgraduate Studies, Research, Technology Transfer and Consultancy (DPRTC) at Sokoine University of Agriculture. Wildlife sampling permits and site access for *C. minuta* were granted by the departments of tourism, and wildlife of Kajiado and Baringo county governments. Approval for environmental (water) collection were similarly obtained from the relevant departments of water, environment and natural resources within both county governments. All bird procedures adhered to international animal welfare standards. No invasive, procedures, anesthesia, or euthanasia were employed, and all birds were released unharmed at the site of capture immediately following processing.

## Results

### Identification and characterization of bacterial isolates

A total of 184 fresh fecal or cloacal samples were collected from *C. minuta* at Lake Bogoria (cohort 1, n = 45; cohort 2, n = 47) and Lake Magadi (cohort 1, n = 36; cohort 2, n = 56). Additionally, 48 water samples were collected from various sites at Lake Bogoria (cohort 1, n = 13; cohort 2, n = 10) and Lake Magadi (cohort 1, n = 13; cohort 2, n = 12) as shown in S1 Table. From these samples, 294 bacterial isolates encompassing 16 genera and 33 distinct species were recovered. Fecal samples yielded 233 isolates (76 from Bogoria, 157 from Magadi), while water samples provided 61 isolates (34 from Bogoria, 27 from Magadi) (S2 Table). A notable similarity was observed in the diversity of Enterobacterales between bird and water samples at both sites. The most frequently isolated species were Enterobacter species (31.0%, n = 91) and *Escherichia coli* (17.3%, n = 51). Beyond the primary Enterobacterales, isolates belonging to the orders Pseudomonadales (*Acinetobacter soli*, *A. hemolyticus*, and *A. balylyi*), Aeromonadales (*Aeromonas veronii* and *A. hydrophila*), and Vibrionales (*Vibrio metschnikovii* and *V. albensis*) were identified.

### Frequency and distribution of MDR and ESBL phenotypes

Of the 294 bacterial isolates 38 (12.9%) were classified as MDR and 19 (6.5%) as ESBL producers only (non-MDR). Additionally, 37 (12.6%) of the isolates co-expressed the ESBL-MDR phenotypes (S2 Table). *Enterobacter* species (8.2%, 24) displayed the highest frequency of MDR. The highest rates of ESBL production was observed in *Acinetobacter* species (3.4%, n = 10). Furthermore, the ESBL-MDR phenotype was most prevalent in *Enterobacter* species (4.8%, n = 14) followed by *Klebsiella* species (2.4%, n = 7).

A notable finding was the frequency of resistance among non-Enterobacterales, specifically *Acinetobacter* species as illustrated in S2 Table.

### Antimicrobial resistance profiles

Resistance profiles varied significantly across study areas, sample sources and cohorts (S2 Table). As shown in Table 1, the highest resistance frequencies were observed for ampicillin (50.0%), amoxicillin-clavulanic acid (36.4%) and tetracycline (32.7%). Moderate resistance was reported in sulfamethoxazole-trimethoprim (24.1%) for both study areas, sample sources and cohorts. In contrast, resistance remained low for several critical antimicrobial classes, including fluoroquinolones (6.5%), aminoglycosides (4.4%), and carbapenems (1.0%). Notably, a disparity was observed among third-generation cephalosporins, with resistance to cefotaxime (18.4%) being substantially higher than resistance to ceftriaxone (5.8%).

### Statistical analysis of resistance distribution

A tie-corrected Kruskal-Wallis test was used to compare median resistance frequencies across study areas, sample sources, and cohorts (Table 2). There were no significant differences in median resistance rates and statistical variations (p values) between *C. minuta* and water isolates, between Lake Bogoria and Lake Magadi or cohorts 1 and 2. However, highly significant differences were observed across the 12 tested antimicrobials. These results indicate that the specific antimicrobial agent was the primary factor associated with variations in resistance, with ampicillin, amoxicillin-clavulanic acid, and tetracycline exhibiting the highest resistance levels, while meropenem maintained the lowest resistance rate.

## Discussion

Our findings align with established research linking anthropogenic pollution to the occurrence of AMR in wild birds. Numerous studies have documented this connection across diverse human-influenced contexts such as intensive agriculture [12,27], landfills and dump sites [18,31], urban areas [32], industries [33], wastewater treatment systems [34,35]. Our

**Table 1. Antimicrobial resistance profiles of isolates (N = 294) categorized by antimicrobial class and agent.**

| Antimicrobial class | Antimicrobial agent | Resistance levels, % (n) |
|---|---|---|
| Penicillins | Ampicillin | 50.0% (147) |
| β-lactam inhibitors | Amoxicillin-clavulanic acid | 36.4% (107) |
| Tetracyclines | Tetracycline | 32.7% (96) |
| Sulfonamides | Sulfamethoxazole-trimethoprim | 24.1% (71) |
| Cephalosporins | Cefotaxime (third-generation) | 18.4% (54) |
| | Ceftriaxone (third-generation) | 5.8% (17) |
| | Cefuroxime (second-generation) | 4.8% (14) |
| | Cefepime (fourth-generation) | 3.4% (10) |
| Fluoroquinolones | Ciprofloxacin | 6.5% (19) |
| Amphenicols | Chloramphenicol | 4.8% (14) |
| Aminoglycosides | Gentamicin | 4.4% (13) |
| Carbapenems | Meropenem | 1.0% (3) |

**Key**: n represents the number of resistant isolates out of N = 294 for all bacterial isolates tested against all antimicrobial agents. Resistance was determined according to CLSI 2024 breakpoints.

**Table 2. Resistance frequencies of bacterial isolates: Descriptive statistics and comparison of variables using a Tie-corrected Kruskal-Wallis Test.**

| Variable | N | Median (IQR) | Min-Max | Test Statistic (df) | P-value |
|---|---|---|---|---|---|
| **Sample sources** | | | | | |
| *C.minuta* | 48 | 8.00(14.85) | 0.00-74.80 | $\chi^2$ (1) = 2.105 | 0.1468 |
| Water | 48 | 5.90(25.00) | 0.00-52.90 | | |
| **Antimicrobials** | | | | | |
| AMC | 8 | 24.40(22.95) | 0.00-60.10 | | |
| AMP | 8 | 35.90(17.70) | 25.80-74.80 | | |
| C | 8 | 0.00(6.3) | 0.00-11.10 | | |
| CIP | 8 | 2.40(13.20) | 0.00-21.40 | | |
| CN | 8 | 4.70(3.50) | 0.00-14.30 | | |
| CRO | 8 | 6.90(9.75) | 0.00-16.10 | | |
| CTX | 8 | 17.70(13.60) | 5.90-35.30 | | |
| CXM | 8 | 4.25(8.50) | 0.00-12.90 | | |
| FEP | 8 | 0.90(6.90) | 0.00-12.90 | | |
| MEM | 8 | 0.00(3.25) | 0.00-7.10 | | |
| SXT | 8 | 10.20(28.10) | 0.00-46.10 | | |
| TE | 8 | 20.20(32.10) | 6.50-51.10 | | |
| **Study areas** | | | | | |
| L. Bogoria | 48 | 6.70(15.85) | 0.00- 52.90 | $\chi^2$ (1) = 0.045 | 0.8325 |
| L. Magadi | 48 | 7.10(25.00) | 0.00−74.80 | | |
| **Cohorts** | | | | | |
| Cohort 1 | 48 | 6.50(16.40) | 0.00−38.50 | $\chi^2$ (1) = 1.573 | 0.2098 |
| Cohort 2 | 48 | 7.10(23.20) | 0.00−74.80 | | |

**Key**: N-counts, IQR-Interquartile Range, df-degrees of freedom, P-value indicates statistical significance. AMC-amoxicillin-clavulanic acid, AMP-ampicillin, C-chloramphenicol, CIP-ciprofloxacin, CN-gentamicin, CRO-ceftriaxone, CTX-cefotaxime, CXM-cefuroxime, FEP-cefepime, MEM-meropenem, SXT-sulfamethoxazole-trimethoprim, TE-tetracycline.

findings confirm that anthropogenic pollution drives AMR occurrence in wild birds. The elevated resistance in *C. minuta* at Lake Magadi likely due to the convergence of high human density, industrial activity, and frequent use of hot springs; which collectively increase the resistant bacterial load at foraging shores. The discharge of industrial and organic waste (from open defecation in the hot springs) acts as nutrient catalysts. This promotes the formation of biofilms, which serve as stable environmental reservoirs that shield resistant bacteria from stressors and facilitate horizontal gene transfer (HGT) among microbial communities. However, the detection of resistant strains at the relatively isolated Lake Bogoria suggests AMR is now embedded in natural ecosystems, allowing migratory birds to acquire resistance far from primary contamination sources.

Aquatic environments are significant conduits for the transmission of resistant bacteria to wild birds [35]. At Bogoria and Magadi sites, freshwater sources serve as critical One Health conduits where overlapping use by humans, livestock, and wildlife facilitates AMR exchange. This process is driven by antimicrobial residues and constant deposits of resistant bacteria from livestock, wildlife or anthropogenic runoff, which provide a steady supply of genetic material. Crucially, these conditions, facilitate HGT primarily through conjugation, enabling the exchange of resistance-conferring plasmids between human-derived pathogens and the indigenous flora of wild birds. This pathway is evidenced by the significantly higher levels of ESBL-producing and MDR Enterobacterales in freshwater sources than in the alkaline lakes, where high salinity limits bacterial viability. The recovery of human-associated pathogens such as, *Leclercia adecarboxylata* [36], *Kluyvera georgiana* [37], multidrug-resistant *Acinetobacter* spp. and high-priority ESKAPE pathogens (*K. pneumoniae* and *Enterobacter* spp.) solidifies the AMR-One Health matrix. These findings expand the study's scope beyond routine environmental monitoring toward a clinical risk framework, highlighting the role of *C. minuta* as a vital sentinel for emerging global antimicrobial threats.

The higher AMR frequency in Cohort 2 birds (pre-departure from the lakes) compared to the arriving Arctic cohort (Cohort 1) suggests resistance is a cumulative process. While Arctic breeding grounds remain ecologically isolated, the Kenyan Rift Valley acts as a high-pressure site for AMR acquisition. This acquisition is likely driven by exposure to polluted water sources under high AMR selection pressure. Alternatively, colonization may occur at high-frequency AMR stopover sites during migration.

The high frequency of isolates resistant to ampicillin, amoxicillin-clavulanic acid, tetracycline, and sulfamethoxazole-trimethoprim antimicrobials heavily used in human and veterinary medicine, suggests environmental contamination driven by the overuse and misuse of these agents. The significant resistance to the WHO-CIA List for human medicine; cefotaxime, ceftriaxone, and ciprofloxacin, is alarming. Furthermore, the presence of ESBL-producing strains in *C. minuta* and water samples is of great concern. This is exacerbated by the location of ESBL genes on mobile genetic elements (MGEs) facilitating their rapid horizontal transfer across bacterial species and ecosystems (14), accelerating the proliferation of ESBLs within the One Health framework [38–40]. The concurrent isolation of MDR strains further narrows the spectrum of effective treatments. While resistance to last-resort antimicrobials like cefepime and meropenem remains low, ongoing global misuse risks fostering resistance to these final-line agents in wild bird populations.

Despite the insights gained, several limitations warrant consideration. The insignificant difference in AMR frequency between cohorts suggests a need for increased sample sizes to validate accumulation trends. Additionally, while phenotypic profiles were established, the absence of whole genomes sequencing (WGS) precludes definitive source-tracing of resistance determinants. Geographically, the focus on the Kenyan Rift Valley captures only a portion of the migratory route; broader surveillance across various stopover sites, paired with longitudinal monitoring, would better elucidate the cumulative nature of AMR acquisition during annual migrations.

## Conclusion

Our findings confirm that aquatic environments at Lake Bogoria and Lake Magadi serve as significant conduits for the transmission of resistant bacteria to wild birds. At these sites, freshwater sources act as critical One Health interfaces

where overlapping use by humans, livestock, and wildlife facilitates the exchange of antimicrobial resistance. The mechanism of this exchange is driven by a steady supply of genetic material from industrial waste, livestock runoff, and open defecation. Crucially, these conditions facilitate HGT, primarily via conjugation, enabling the exchange of resistance-conferring plasmids between human-derived pathogens and the indigenous flora of *C. minuta*. Consequently, this places *C. minuta* as a critical sentinel and potential accelerator for the environmental spread of high-priority pathogens, including MDR and ESBL-producing strains and resistant ESKAPE group. Since wild birds are not direct recipients of antimicrobial therapies, their role in the AMR cycle necessitates a paradigm shift: transitioning from clinical-centric models to proactive surveillance within a holistic One Health framework. To fully address these risks, global surveillance must integrate longitudinal monitoring and genomic analysis to track the movement of mobile genetic elements across diverse landscapes.

## Supporting information

**S1 Table. Summary of fecal samples from *C. minuta* and water samples collected from Lake Bogoria and Lake Magadi.**
(XLSX)

**S2 Table. Antimicrobial resistance profiles of bacterial isolates from *C. minuta* fecal samples and water samples from Lake Bogoria and Lake Magadi.**
(XLSX)

## Acknowledgments

We acknowledge and express gratitude to the Center for Microbiology Research at the Kenya Medical Research Institute (CMR-KEMRI), SACIDS Foundation for One Health at the Sokoine University of Agriculture, the International Livestock Research Institute and Karatina University for guidance, infrastructure, and laboratory supplies. We also specifically thank Gerishom Angote, Benard Ochiel, and Hulder Otieno at CMR-KEMRI for their immense laboratory input.

## Author contributions

**Conceptualization:** CATHERINE W. MBUTHIA, Titus S. Imboma, John Kiiru, Abubakar S. Hoza.

**Data curation:** CATHERINE W. MBUTHIA, Rael J. Too, John Kiiru, Abubakar S. Hoza.

**Formal analysis:** CATHERINE W. MBUTHIA.

**Funding acquisition:** CATHERINE W. MBUTHIA.

**Investigation:** CATHERINE W. MBUTHIA, Titus S. Imboma.

**Methodology:** CATHERINE W. MBUTHIA, Rael J. Too.

**Project administration:** CATHERINE W. MBUTHIA, Abubakar S. Hoza.

**Resources:** Rael J. Too, John Kiiru, Samuel Kariuki.

**Supervision:** Alexanda Mzula, Samuel Kariuki, Abubakar S. Hoza.

**Validation:** CATHERINE W. MBUTHIA, Rael J. Too, John Kiiru, Samuel Kariuki, Abubakar S. Hoza.

**Visualization:** CATHERINE W. MBUTHIA, Rael J. Too, John Kiiru, Samuel Kariuki, Abubakar S. Hoza.

**Writing – original draft:** CATHERINE W. MBUTHIA.

**Writing – review & editing:** CATHERINE W. MBUTHIA, Rael J. Too, Alexanda Mzula, John Kiiru, Samuel Kariuki, Abubakar S. Hoza.

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
