## [Decision Letter · Decision Letter 0]

23 Nov 2025

Dear Dr. MBUTHIA,

Thank you for submitting your manuscript to PLOS ONE. After careful consideration, we feel that it has merit but does not fully meet PLOS ONE’s publication criteria as it currently stands. Therefore, we invite you to submit a revised version of the manuscript that addresses the points raised during the review process.

We look forward to receiving your revised manuscript.

Kind regards,

Mabel Kamweli Aworh, DVM, MPH, PhD. FCVSN

Academic Editor

PLOS ONE

Journal Requirements:

3. We note that Figures 1A and 1B in your submission contain map images which may be copyrighted. All PLOS content is published under the Creative Commons Attribution License (CC BY 4.0), which means that the manuscript, images, and Supporting Information files will be freely available online, and any third party is permitted to access, download, copy, distribute, and use these materials in any way, even commercially, with proper attribution. For these reasons, we cannot publish previously copyrighted maps or satellite images created using proprietary data, such as Google software (Google Maps, Street View, and Earth). For more information, see our copyright guidelines: http://journals.plos.org/plosone/s/licenses-and-copyright ..

1. You may seek permission from the original copyright holder of Figures 1A and 1B to publish the content specifically under the CC BY 4.0 license.

Additional Editor Comments:

In addition to addressing the reviewers comments, please highlight key limitations of the study as the las paragraph of the Discussion section. Also ensure that 80% of the reference list is within the last five years.

Reviewers' comments:

Reviewer's Responses to Questions

**Comments to the Author**

1. Is the manuscript technically sound, and do the data support the conclusions?

Reviewer #1: Yes

Reviewer #2: Yes

2. Has the statistical analysis been performed appropriately and rigorously?

Reviewer #1: Yes

Reviewer #2: I Don't Know

3. Have the authors made all data underlying the findings in their manuscript fully available?

Reviewer #1: Yes

Reviewer #2: Yes

4. Is the manuscript presented in an intelligible fashion and written in standard English?

Reviewer #1: Yes

Reviewer #2: Yes

Reviewer #1: General Comment

This paper offers truly important insights into how migratory birds play a role in spreading antimicrobial resistance. Its findings are vital for guiding public health policies and environmental strategies against AMR. What's particularly exciting is that this study uncovers a link between the little stint bird and AMR, thoroughly addressing a big gap in what we currently know. Looking ahead, authors should consider sequencing their isolates, as a future study. It would provide much deeper genomic insights, especially into the mobile genetic elements that carry or share these resistance genes.

Specific Comments

Line 242–243: The table layout appears cluttered, making it hard to read, especially at publication scale. Also, consider defining these abbreviations (for example “C1”, “C2”) mentioned in the caption.

Line 256: Insert a space between “1” and “and.”

Lines 261–262: Similar to Table 1, define all abbreviations for clarity. Improve table formatting for readability.

Lines 274–278: Good you mentioned, potential AMR factors such as human interactions at hot springs, rivers shared with livestock, and salt pans effluent. Consider elaborating on how these factors directly contribute to AMR in migratory birds in the regions. Expand the discussion to explain the mechanisms or pathways through which these factors may directly or indirectly influence AMR transmission or persistence.

Lines 442–443: Review reference formatting (for instance citations, 35 and 39) and ensure consistent use of the appropriate citation style as required by the journal throughout the reference list.

Reviewer #2: The topic of the study is definitely of interest to the AMR community. Minor corrections are suggested as enumerated below

1. Paragraph starting at line 64. Introducing a statement or 2 describing how the little stint (Calidris minuta) specifically feeds and generally interacts with the niche upon arrival at lakes Magadi and Bogoria will be informative.

2. Line 77. "While anthropogenic activities are recognized as a key driver of AMR dissemination by migratory birds globally,", kindly provide reference for this.

3. Line 94. "Anthropogenic activity is substantial due to the commercial trona and soda ash mining industry." Knowing that PLoS One is a generalist publication, the authors might wish to clarify to readers how the commercial trona might contribute to AMR risk at this site.

4. Line 90. "Human interactions with the lake are minimal, with the agro-pastoralists Tugens and animals (domestic and wildlife) accessing the freshwater springs by the lakes." At present, I am not convinced, as per this statement, that the human interaction due to the agro-pastoralists and their animals are minimal (no evidence provided) or minimal compared to lake Magadi. The authors should provide clarifying evidence.

5. Line 117. Reporting the results of number of samples collected under the methods section is inappropriate. Such data should be in the result section.

6. Line 120. We are told that water samples were drawn from lakes, hot springs, rivers, and springs [4 sources] used by the little stint. What we are not told is whether the 48 samples that was eventually collected was achieved using random or other form of sampling from these 4 sources. Please clarify.

7. At line 153, "They included; ampicillin (AMP, 10 μg), amoxicillin-clavulanic acid (AMC, 30 (20:10) μg), cefepime (FEP, 30 μg), cefotaxime (CTX, 30 μg), ceftriaxone (CRO, 30 μg), cefuroxime (CXM, 30 μg), meropenem (MEM, 10 μg), ciprofloxacin (CIP, 5 μg), tetracycline (TE, 30 μg), chloramphenicol (C, 30 μg), gentamicin (CN, 10 μg), trimethoprim sulfamethoxazole (SXT, 25 (1.25: 23.75). The authors, might explain to PONE readers what the numbers with the μg units mean and why μg is missing for trimethoprim sulfamethoxazole.

8. Full legend information should accompany each table. For example, the authors should not leave readers to assume exactly what C1/C2 means. In Table 1, the value of the data within and outside brackets in each cell should be defined at the top of the column.

9. The authors should introduce a section on limitations encountered in this study. Limitations include lack of molecular studies to achieve source attribution for the bacteria isolated as well as determine dissemination pathways. Though the authors have tried to indirectly and subtly link the Rift Valley niche to AMR pathogens in these birds by mentioning environmental stewardship. Evidence for such is not presented in this study.

**Do you want your identity to be public for this peer review?** For information about this choice, including consent withdrawal, please see our For information about this choice, including consent withdrawal, please see our Privacy Policy .

Reviewer #1: No

Reviewer #2: No

---

## [Author Response · Author response to Decision Letter 1]

15 Feb 2026

2. We note that your Data Availability Statement is currently as follows: [All relevant data are within the manuscript and its Supporting Information files- I have added other supporting information files (S1 and S2 Tables that have all the sample descriptives and AMR profiles of all isolates respectively, for all the study areas, sample sources and cohorts.

Please confirm at this time whether or not your submission contains all raw data required to replicate the results of your study. Authors must share the “minimal data set” for their submission. PLOS defines the minimal data set to consist of the data required to replicate all study findings reported in the article, as well as related metadata and methods -(https://journals.plos.org/plosone/s/data-availability#loc-minimal-data-set-definition)-Added

- The points extracted from images for analysis- Attached in the Supporting information as S1 Table, S2 Table and Tie-corrected Kruskal-Wallis Test Analysis

If your submission does not contain these data, please either upload them as Supporting Information files or deposit them to a stable, public repository and provide us with the relevant URLs, DOIs, or accession numbers. For a list of recommended repositories, please see https://journals.plos.org/plosone/s/recommended-repositories-Attached in the Supporting information

3. We note that Figures 1A and 1B in your submission contain map images which may be copyrighted. All PLOS content is published under the Creative Commons Attribution License (CC BY 4.0), which means that the manuscript, images, and Supporting Information files will be freely available online, and any third party is permitted to access, download, copy, distribute, and use these materials in any way, even commercially, with proper attribution. For these reasons, we cannot publish previously copyrighted maps or satellite images created using proprietary data, such as Google software (Google Maps, Street View, and Earth). For more information, see our copyright guidelines: http://journals.plos.org/plosone/s/licenses-and-copyright.

1. You may seek permission from the original copyright holder of Figures 1A and 1B to publish the content specifically under the CC BY 4.0 license.

2. If you are unable to obtain permission from the original copyright holder to publish these figures under the CC BY 4.0 license or if the copyright holder’s requirements are incompatible with the CC BY 4.0 license, please either i) remove the figure or ii) supply a replacement figure that complies with the CC BY 4.0 license. Please check copyright information on all replacement figures and update the figure caption with source information. If applicable, please specify in the figure caption text when a figure is similar but not identical to the original image and is therefore for illustrative purposes only

The maps (Figure 1 and Figure 2) were generated using QGIS software version 3.4, which is an open-source Geographic Information System licensed under the GNU General Public License (GPL). As per the QGIS Foundation, the software creators do not hold copyright over works produced by users. Therefore, as the author of the maps, I grant full permission for them to be published under the CC BY 4.0 license.

I have updated the figure captions within the manuscript to explicitly state the open-access data sources used (OpenStreetMap and HDX), ensuring that no copyrighted proprietary base maps (such as Google Maps or Esri) were utilized.

Updated Captions: ‘Maps were created by the authors using QGIS version 3.4. Administrative boundaries are sourced from the Humanitarian Data Exchange (licensed under CC BY 4.0). Lake boundaries and river networks are sourced from OpenStreetMap (available under the Open Database License). All other data points were collected by the authors. This figure is original and licensed under CC BY 4.0.’

Please review your reference list to ensure that it is complete and correct. If you have cited papers that have been retracted, please include the rationale for doing so in the manuscript text, or remove these references and replace them with relevant current references. Any changes to the reference list should be mentioned in the rebuttal letter that accompanies your revised manuscript. If you need to cite a retracted article, indicate the article’s retracted status in the References list and also include a citation and full reference for the retraction notice-I have not cited any retracted paper

Additional Editor Comments:

In addition to addressing the reviewers comments, please highlight key limitations of the study as the last paragraph of the Discussion section. Also ensure that 80% of the reference list is within the last five years. – Included in Lines 311-318

REVIEWERS' COMMENTS:

Reviewer's Responses to Questions

Comments to the Author

1. Is the manuscript technically sound, and do the data support the conclusions?

Reviewer #1: Yes

Reviewer #2: Yes

2. Has the statistical analysis been performed appropriately and rigorously?

Reviewer #1: Yes

Reviewer #2: I Don't Know

3. Have the authors made all data underlying the findings in their manuscript fully available?

Reviewer #1: Yes

Reviewer #2: Yes

4. Is the manuscript presented in an intelligible fashion and written in standard English?

Reviewer #1: Yes

Reviewer #2: Yes

5. Review Comments to the Author

Reviewer #1: General Comment

This paper offers truly important insights into how migratory birds play a role in spreading antimicrobial resistance. Its findings are vital for guiding public health policies and environmental strategies against AMR. What's particularly exciting is that this study uncovers a link between the little stint bird and AMR, thoroughly addressing a big gap in what we currently know. Looking ahead, authors should consider sequencing their isolates, as a future study. It would provide much deeper genomic insights, especially into the mobile genetic elements that carry or share these resistance genes.

Specific Comments

Line 242–243: The table layout appears cluttered, making it hard to read, especially at publication scale. Also, consider defining these abbreviations (for example “C1”, “C2”) mentioned in the caption. Did an overhaul of Table 1 and C1 and C2 -cohorts 1 and 2 are no longer included- Lines 244-247

Line 256: Insert a space between “1” and 2. Rewritten the paragraph-Lines 250-257

Lines 261–262: Similar to Table 1, define all abbreviations for clarity. Improve table formatting for readability-Done lines 260-263 under key

Lines 274–278: Good you mentioned, potential AMR factors such as human interactions at hot springs, rivers shared with livestock, and salt pans effluent. Consider elaborating on how these factors directly contribute to AMR in migratory birds in the regions. Expand the discussion to explain the mechanisms or pathways through which these factors may directly or indirectly influence AMR transmission or persistence-Rephrased at lines 281-287

Lines 442–443: Review reference formatting (for instance citations, 35 and 39) and ensure consistent use of the appropriate citation style as required by the journal throughout the reference list-Removed Ref 39 but fixed Ref 35

Reviewer #2: The topic of the study is definitely of interest to the AMR community. Minor corrections are suggested as enumerated below

1. Paragraph starting at line 64. Introducing a statement or 2 describing how the little stint (Calidris minuta) specifically feeds and generally interacts with the niche upon arrival at lakes Magadi and Bogoria will be informative- Corrected at lines 67-71.

2. Line 77. "While anthropogenic activities are recognized as a key driver of AMR dissemination by migratory birds globally,", kindly provide reference for this- Done-Line 83

3. Line 94. "Anthropogenic activity is substantial due to the commercial trona and soda ash mining industry." Knowing that PLoS One is a generalist publication, the authors might wish to clarify to readers how the commercial trona might contribute to AMR risk at this site-This is explicitly explained in the discussion (Line 270-276), and Lines 281-287 rather than in the study area description

4. Line 99. "Human interactions with the lake are minimal, with the agro-pastoralists Tugens and animals (domestic and wildlife) accessing the freshwater springs by the lakes." At present, I am not convinced, as per this statement, that the human interaction due to the agro-pastoralists and their animals are minimal (no evidence provided) or minimal compared to lake Magadi. The authors should provide clarifying evidence. Paraphrased at line 107-110. Since the main bodies of both lakes are unsuitable for hydration or domestic use due to their extreme chemistry, significant interaction between humans, livestock, and C. minuta occurs at the surrounding peripheral freshwater sources, including springs, rivers, marshes, and boreholes.

5. Line 117. Reporting the results of number of samples collected under the methods section is inappropriate. Such data should be in the result section. Corrected, now at Lines 211-214

6. Line 120. We are told that water samples were drawn from lakes, hot springs, rivers, and springs [4 sources] used by the little stint. What we are not told is whether the 48 samples that was eventually collected was achieved using random or other form of sampling from these 4 sources. Please clarify. Corrected-Line 137- Water samples were collected using criterion purposive sampling

7. At line 153, "They included; ampicillin (AMP, 10 μg), amoxicillin-clavulanic acid (AMC, 30 (20:10) μg), cefepime (FEP, 30 μg), cefotaxime (CTX, 30 μg), ceftriaxone (CRO, 30 μg), cefuroxime (CXM, 30 μg), meropenem (MEM, 10 μg), ciprofloxacin (CIP, 5 μg), tetracycline (TE, 30 μg), chloramphenicol (C, 30 μg), gentamicin (CN, 10 μg), trimethoprim sulfamethoxazole (SXT, 25 (1.25: 23.75). The authors, might explain to PONE readers what the numbers with the μg units mean and why μg is missing for trimethoprim sulfamethoxazole-Corrected at lines 172 and 177 A panel of 12 antimicrobial disks (Oxoid, Basingstoke, UK), representing 8 drug classes of varying concentrations (μg)….. sulfamethoxazole-trimethoprim (SXT, 25 μg).

8. Full legend information should accompany each table. For example, the authors should not leave readers to assume exactly what C1/C2 means. In Table 1, the value of the data within and outside brackets in each cell should be defined at the top of the column-Edited Table 1 (cohort is no longer part of this table) and written full cohort in Table 2

9. The authors should introduce a section on limitations encountered in this study. Limitations include lack of molecular studies to achieve source attribution for the bacteria isolated as well as determine dissemination pathways. Though the authors have tried to indirectly and subtly link the Rift Valley niche to AMR pathogens in these birds by mentioning environmental stewardship. Evidence for such is not presented in this study- Lines 316-323

---

## [Decision Letter · Decision Letter 1]

11 Mar 2026

A One Health Assessment of Antimicrobial-Resistant Enterobacterales in Migratory Little Stints (Calidris minuta) and Aquatic Ecosystems in the Kenyan Rift Valley

PONE-D-25-57615R1

Dear Dr. Mbuthia,

We’re pleased to inform you that your manuscript has been judged scientifically suitable for publication and will be formally accepted for publication once it meets all outstanding technical requirements.

Kind regards,

Mabel Kamweli Aworh, DVM, MPH, PhD. FCVSN

Academic Editor

PLOS One

Additional Editor Comments (optional):

Reviewers' comments:

Reviewer's Responses to Questions

**Comments to the Author**

Reviewer #1: All comments have been addressed

Reviewer #2: All comments have been addressed

2. Is the manuscript technically sound, and do the data support the conclusions?

Reviewer #1: Yes

Reviewer #2: (No Response)

3. Has the statistical analysis been performed appropriately and rigorously?

Reviewer #1: Yes

Reviewer #2: (No Response)

4. Have the authors made all data underlying the findings in their manuscript fully available?

Reviewer #1: Yes

Reviewer #2: (No Response)

5. Is the manuscript presented in an intelligible fashion and written in standard English?

Reviewer #1: Yes

Reviewer #2: (No Response)

Reviewer #1: Thank you for addressing my comments. A minor correction is required on lines 31–32 of the abstract, as content appears to be omitted between "lowest for" and "and meropenem."

Reviewer #2: "Since the main bodies of both lakes are unsuitable for hydration or domestic use due to their extreme chemistry, significant interaction between humans, livestock, and C. minuta occurs at the surrounding peripheral freshwater sources, including springs, rivers, marshes, and boreholes." This new statement added at Line 107 is welcomed. However, it might require a reference especially if the authors did not specifically observe what was stated.

**Do you want your identity to be public for this peer review?** For information about this choice, including consent withdrawal, please see our For information about this choice, including consent withdrawal, please see our Privacy Policy .

Reviewer #1: No

Reviewer #2: No

---

## [Editor Report · Acceptance letter]

PONE-D-25-57615R1

PLOS One

Dear Dr. MBUTHIA,

I'm pleased to inform you that your manuscript has been deemed suitable for publication in PLOS One. Congratulations! Your manuscript is now being handed over to our production team.

Kind regards,

on behalf of

Dr. Mabel Kamweli Aworh

Academic Editor

PLOS One